# Differential Regulation of Circadian Clock Genes by UV-B Radiation and 1,25-Dihydroxyvitamin D: A Pilot Study during Different Stages of Skin Photocarcinogenesis

**DOI:** 10.3390/nu16020254

**Published:** 2024-01-14

**Authors:** Leandros Lamnis, Christoforos Christofi, Alexandra Stark, Heike Palm, Klaus Roemer, Thomas Vogt, Jörg Reichrath

**Affiliations:** 1Dermatology, University of Saarland Medical Center, 66421 Homburg, Germany; leandroslamnis@gmail.com (L.L.); thomas.vogt@uks.eu (T.V.); 2José Carreras Center and Internal Medicine I, 66421 Homburg, Germany

**Keywords:** circadian clock, circadian rhythmicity, vitamin D, vitamin D signaling, vitamin D receptor, vitamin D receptor signaling, skin, photocarcinogenesis, skin photocarcinogenesis, skin cancer, ultraviolet radiation

## Abstract

Background: Increasing evidence points at an important physiological role of the timekeeping system, known as the circadian clock (CC), regulating not only our sleep–awake rhythm but additionally many other cellular processes in peripheral tissues. It was shown in various cell types that environmental stressors, including ultraviolet B radiation (UV-B), modulate the expression of genes that regulate the CC (CCGs) and that these CCGs modulate susceptibility for UV-B-induced cellular damage. It was the aim of this pilot study to gain further insights into the CCs’ putative role for UV-B-induced photocarcinogenesis of skin cancer. Methods: Applying RT-PCR, we analyzed the expression of two core CCGs (brain and muscle ARNT-like 1 (Bmal1) and Period-2 (Per2)) over several time points (0–60 h) in HaCaT cells with and without 1,25-dihydroxyvitamin D (D_3_) and/or UV-B and conducted a cosinor analysis to evaluate the effects of those conditions on the circadian rhythm and an extended mixed-effects linear modeling to account for both fixed effects of experimental conditions and random inter-individual variability. Next, we investigated the expression of these two genes in keratinocytes representing different stages of skin photocarcinogenesis, comparing normal (Normal Human Epidermal Keratinocytes—NHEK; p53 wild type), precancerous (HaCaT keratinocytes; mutated p53 status), and malignant (Squamous Cell Carcinoma SCL-1; p53 null status) keratinocytes after 12 h under the same conditions. Results: We demonstrated that in HaCaT cells, Bmal1 showed a robust circadian rhythm, while the evidence for Per2 was limited. Overall expression of both genes, but especially for Bmal1, was increased following UV-B treatment, while Per2 showed a suppressed overall expression following D_3_. Both UVB and 1,25(OH)_2_D_3_ suggested a significant phase shift for Bmal1 (*p* < 0.05 for the acrophase), while no specific effect on the amplitude could be evidenced. Differential effects on the expression of BMAL1 and Per2 were found when we compared different treatment modalities (UV-B and/or D_3_) or cell types (NHEK, HaCaT, and SCL-1 cells). Conclusions: Comparing epidermal keratinocytes representing different stages of skin photocarcinogenesis, we provide further evidence for an independently operating timekeeping system in human skin, which is regulated by UV-B and disturbed during skin photocarcinogenesis. Our finding that this pattern of circadian rhythm was differentially altered by treatment with UV-B, as compared with treatment with D_3_, does not support the hypothesis that the expression of these CCGs may be regulated via UV-B-induced synthesis of vitamin D but might be introducing a novel photoprotective property of vitamin D through the circadian clock.

## 1. Introduction

The first ever documentation of a genetic element encoding biologic timing can be traced back to the works of Ronald Konopka and Seymour Benzer on the common fruit fly in the 1970s [1]. The successful identification, afterward, of the period (PER) gene in 1984 [2] and the further illumination of its timekeeping properties [3,4] by the teams of Jeffrey Hall, Michael Rosbash, and Michael Young granted them the 2017 Nobel in Physiology and Medicine for setting the bases for circadian physiology research. In the time since, research of the circadian clock has come a very long way. The term “circadian” originates from the Latin words circa (“around”) and diēm (“day”); therefore, the circadian rhythm represents a 24 h circle of day and night, and biological circadian clocks entail those processes through which our body responds to the constant rotation of the Earth around its axis. The genes involved in the control of biologic timekeeping are thus called circadian clock genes (CCGs) [5].

Circadian clocks exist in almost all human cells and mediate their timekeeping properties through a complex system of autoregulatory transcriptional/translational feedback loops (TTFLs), coordinated under a master pacemaker in the suprachiasmatic nucleus (SCN) of the hypothalamus. External stimuli called “Zeitgeber” (time cues) entrain the rhythm of the SCN, which through a series of electrical and molecular circadian messages synchronizes subordinate peripheral clocks in other tissues [6,7]. Light has been suggested as the most dominant “Zeitgeber”, signaling the constant alterations of daily light–dark circles to the SCN through the retino-hypothalamic tract. Feeding–fasting circles have also been shown to be heavily involved in the synchronization of the central circadian clock, and the circadian physiology of metabolism has recently become a center of research attention [8]. However, the SCN is also capable of maintaining circadian rhythmicity even in the absence of external stimulation. Moreover, despite the strict hierarchical model of circadian clock synchronization having been popular for a long time, the discovery of peripheral clocks in multiple tissues, including the skin, which interact with time cues, like light, independently of the central clock, suggests a more complex system of clock organization [9].

While tissue-specific differences exist between different clocks, a basic layout consisting of “core clock genes” and their respective interaction patterns with one another (as part of the TTFL) is common among all circadian oscillators. The genes that constitute the “core clock genes” are brain and muscle ARNT-like 1 (Bmal1), circadian locomotor output cycles kaput (Clock), cryptochromes (Cry1 and Cry2), and periods (Per1, Per2, and Per3) [8,9]. BMAL1 and CLOCK proteins bind in the cytoplasm forming a heterodimer that translocates to the cellular nucleus. There it binds to the E-box motif in the promoter regions of “clock-controlled genes”, estimated to involve around 10% of the genome and thereby fostering their expression. At the same time, transcription of the PER and CRY families of proteins is also stimulated. This second heterodimer inhibits the binding of BMAL1 and CLOCK and BMAL1/CLOCK-mediated transcription, thereby obstructing their own formation and closing the feedback loop circle. Protein degradation allows the circle to repeat in approximately 24 h lasting intervals [9,10].

The skin constitutes a very important model for studying the complexities of the circadian clock systems. It involves multiple cell-types structured both across delimited compartments (epidermis-keratinocytes, dermis-fibroblasts, adipose tissue-fat cells) and with interconnected skin cells (melanocytes, hair follicles, sebaceous glands), immune cells (Langerhans cells, T-lymphocytes, mast cells) with distinct but probably coordinated circadian clocks. Moreover, several physiologic cutaneous processes, including responses to environmental stress, have been shown to be modulated by the circadian clock [6,7,10,11,12,13,14].

Notably, some skin cells are in contact with nerval endings, and therefore in probable communication with the SCN, while others are not. Transcellular and intercellular regulation of clock synchronization further gain importance as a promising research field that accounts for the skin’s direct proximity to the external environment.

Skin cancer is the most common form of cancer in humans from which the vast majority represent non-melanoma skin cancer, namely, basal and squamous cell carcinomas (BCC and SCC, respectively). Our skin is daily exposed to a number of environmental insults like pollutants and solar radiation. Solar ultraviolet (UV) radiation has been long known to play a major role in skin photocarcinogenesis, indicated also by the higher rates of skin cancer in outdoor compared with indoor workers [15]. Non-melanoma skin cancer formation induced by UV radiation is characterized by a three-step pathogenetic process: (1) Initiation involves the creation and collection of genetic mutations (photolesions) that affect by extension and alter signal transduction pathways. Failure to repair the initiated damage can lead to (2) promotion, which is characterized by clonal expansion of the cells involved, and subsequently to (3) progression, which refers to the malignant transformation of those cells [16]. Solar UV radiation (UV-R) is divided into three main subtypes, UVA (320–400 nm), UVB (290–320 nm), and UVC (200–290 nm), but only UVA and UVB do in fact reach the surface of Earth after partial absorption by the atmosphere. In spite of it accounting for only a minor part of the UV reaching Earth, UVB represents the major cause of photoaging and UV-induced skin cancer [15].

The two major photoproducts (photolesions) are cyclobutane pyrimidine dimers (CPDs) and 6–4 photoproducts (6–4 PPs), both mediating mutagenic, cytotoxic, and carcinogenic processes, with CPDs being the most abundant ones. Nevertheless, the cells have evolved certain protective and repair mechanisms to counteract those harmful effects of UV-R [17]. The nucleotide excision repair system (NER) constitutes an enzymatic system capable of the detection and removal of DNA damage. After the detection of DNA damage in eukaryotic organisms, the NER implements a dual incision peripheral to the damaged base, releasing it as a 24- to 32-nucleotide-long oligomer, while a polymerase replaces the formed oligomer gap, and a ligase finally seals the repair patch. The NER represents the sole repair system for CPD and 6–4 PP in humans and mice [13,18].

Increasing evidence suggests a direct regulatory relationship between the body’s UV response and the circadian clock. In humans, mutations in the gene encoding for xeroderma pigmentosum group A (XPA), a vital part of the nucleotide excision repair (NER)-repair system, result in xeroderma pigmentosum. This syndrome is linked with a 5000-fold increase in skin cancer incidence in sun-exposed body areas. Interestingly, XPA has recently been found to be controlled by the circadian clock, resulting in a time dependency of UV-induced cellular damage [13]. The time of the day of UV exposure has been linked to sunburn apoptosis, inflammatory cytokine induction, and erythema through time-dependent regulation of the p53 tumor suppressor gene [19]. In essence, multiple sources of evidence indicate that evening radiation, when DNA replication is at its peak and DNA repair “rests”, finds the human skin at its most vulnerable state against the UV radiation’s damaging effects. This phenomenon has been directly linked to several core CCGs like BMAL1, CLOCK [20,21,22], Cry2 [23,24], Per2 [21], etc. As a consequence, several scientists have already proposed that a chemically assisted photo-inducible modification of the circadian clock may pave the way for a new field in medicine termed “chronophotopharmacology” [25], particularly in response to UV radiation.

It was the aim of this pilot study to gain further insights into the CCGs’ putative role for UV-B-induced skin photocarcinogenesis. In particular, we aimed to investigate UV effects on the expression of two major CCGs and whether these effects may at least in part be mediated by 1,25(OH)_2_D_3_ (D_3_), the active form of vitamin D, which depends on UV-B for its cutaneous synthesis and is known to protect the skin from UV-B’s damaging properties.

## 2. Methods

### 2.1. Cellular Models

#### 2.1.1. HaCaT Keratinocytes

Human adult/calcium/temperature (HaCaT) keratinocytes (CLS Cell Lines Service^®^, Eppelheim, Germany) were cultivated in Dulbecco’s modified eagle’s medium (DMEM), supplemented with 1% L-Glutamine and 10% Fetal Bovine Serum (Gibco, Thermo Fisher Scientific, Dreieich, Germany). HaCaT cells are increasingly recognized as a promising tool to study the process of malignant transformation of human epithelial cells [26] and have been recently proven to have a functional cellular autonomous circadian clock [27].

#### 2.1.2. SCL-1 Cells

We used SCL-1 cells (a gift from the German Center of Cancer Research, Heidelberg, Germany). SCL-1 cells represent a poorly differentiated form of cutaneous squamous cell carcinoma, also maintaining their morphological stability over a high number of passages. Cutaneous SCC cell lines often present a null p53 phenotype due to multiple mutations on the tumor suppressor gene [28]. We maintained them in Roswell Park Memorial Institute (RPMI) 1640 medium, supplemented with 1% L-Glutamine and 10% Fetal Bovine Serum (Thermo Fisher Scientific).

#### 2.1.3. NHEK Cells

We used two sets of pooled juvenile Normal Human Epidermal Keratinocytes (NHEK, Promocell, Heidelberg, Germany; Lot. No.: 459Z009 and 466Z002), each coming from three different donors, which were then further pooled together. Overall, our cultures were thereby coming from a total of six different donors. We cultured them in Keratinocyte Growth Medium 2, supplemented with SupplementMix and 0.01% CaCl_2_ Solution (Promocell).

### 2.2. Treatments

UV-B treatment: Following the removal of the culture medium from the cell culture dishes, the cells underwent a washing step using phosphate-buffered saline (PBS) before being exposed to UVB irradiation. The UVB exposure was conducted at a dose of 50 mJ/cm^2^, with a midrange wavelength of 302 nm, utilizing the Crosslinker CL-1000M (Ultra-violet products Ltd., purchased by Analytik Jena, Jena, Germany). Subsequently, the cells were supplemented with fresh medium [and for combination treatments, further including 1,25(OH)_2_D_3_ as described below].

1,25(OH)_2_D_3_ treatment: Cells were treated with 1,25(OH)_2_D_3_ (Sigma, Taufkirchen, Germany) at a final concentration of 10^−7^ M, which was achieved by dissolving an ethanol (EtOH) stock solution (10^−4^ M) at a 1:1000 dilution in medium. To mitigate non-specific binding of 1,25(OH)_2_D_3_ to the dish, bovine serum albumin (BSA, Sigma) at a concentration of 1% was incorporated into the medium during treatment. Vehicle control samples included EtOH at a 1:1000 dilution (5 μL per 5 mL medium per dish) and BSA (1%). Importantly, preliminary experiments were conducted to ascertain that EtOH had no impact on gene expression, as cells treated with BSA alone exhibited comparable results to those treated with BSA and EtOH.

### 2.3. LDH Toxicity Assays

This experiment was performed to obtain further insights into the contribution of two independent vitamin D-dependent signaling pathways that are mediated by the classical vitamin D receptor (VDR) and the aryl hydrocarbon receptor (AhR) for the cellular response to UV-B-induced damage. We applied an AhR-Antagonist CH223191 (Sigma) at a final concentration of 10^−7^ M, achieved by diluting an ethanol stock solution (10^−4^ M) at 1:1000 in medium and a partial VDR-Inhibitor (25-Hydroxyvitamin D_3_ [25(OH)_2_D_3_, Sigma]) at the same final concentration of 10^−7^ M. Previous studies [29,30,31] have confirmed the effective blocking of the AhR and VDR receptors by CH223191 and 25(OH)_2_D_3_, respectively, at this concentration. We used all 16 possible combinations of treatment factors UVB, 1,25(OH)_2_D_3_, CH223191, and 25(OH)_2_D_3_ as illustrated in Table 1. As the control (standard) we used an extra dish treated with Triton-X Solution (Merck Millipore, Burlington, VT, USA) diluted down to 1% in DMEM containing 1% BSA and incubated for 15′. LDH activity was measured with a Cytotoxicity Detection Kit (Roche, Mannheim, Germany). The LDH activity measured in the free medium of treated cells was directly proportional to the level of toxicity undergone by the cells under their respective treatments [32].

### 2.4. Sampling of Cells

HaCaT cells were harvested immediately after treatment (t = 0 h) and thereafter in 6 h increments for a total of 60 h (0 h, 6 h, 12 h, …, 60 h), for a total of 11 measurements per treatment condition. NHEK and SCL-1 were harvested 12 h following treatment. For LDH toxicity assays, the medium was aspirated from HaCaT, 24 h following treatment.

RNA isolation was conducted using an RNeasy Kit and QIA shredder (Qiagen, Hilden, Germany) and cDNA reverse transcription using Omniscript RT Kit (Qiagen). Oligo-dT-primers, RNase inhibitors, and 1 μg mRNA were used in every reaction as templates. Quantitative real-time PCR (RT-qPCR) was performed with the QuantiTect SYBR Green PCR Kit (Qiagen) in 96-well plates with 120 cycles in a StepOnePlus Real-Time PCR System (Thermo Fisher Scientific), using duplicates for each sample. Expression of the target genes was measured with purchased gene-specific primers (Qiagen) for BMAL1/ARNTL (QT00011844), Per2 (QT0001120), and GADPH (QT00079247) as a reference gene. We calculated the relative gene expression (ΔCt) based on this and used it for statistics. ΔΔCt values, representing the expression ratio, were calculated by subtracting the ΔCt of the treatment condition from the ΔCt of the control condition. For multiple time points we used the control condition at the time of treatment (t = 0 h) as the overall control. All data were represented as a mean ± standard deviation (SD) of three experiment replications.

### 2.5. Statistical Analysis

To analyze the temporal expression patterns of Bmal1 and Per2 genes in HaCaT keratinocytes under different treatment conditions, we employed a comprehensive approach using three statistical methods: cosinor analysis using the “cosinor2” (https://cran.r-project.org/web/packages/cosinor2/index.html, accessed on 11 January 2024) package in R programming language (Rstudio, Version 4.3.2), linear mixed-effects modeling (MELM) with the “nlme” (https://cran.r-project.org/web/packages/nlme/index.html, accessed on 11 January 2024) package, and repeated measures ANOVA (in SPSS v26; see below).

#### 2.5.1. Cosinor Analysis with “cosinor2”

The cosinor analysis is a statistical method commonly used in chronobiology to assess rhythmic patterns in time series data. It is particularly suitable for detecting oscillatory patterns, such as circadian rhythms in gene expression. The primary goal of cosinor analysis is to determine whether a periodic component, modeled as a cosine function, significantly contributes to the observed variation in a time series. The key components of interest in cosinor analysis are the amplitude (indicating the strength of the rhythm), acrophase (timing of the peak), and mesor (mean value around which the oscillation occurs) [33,34]. This method fits a sinusoidal model to the data, represented by the formula [35]
$$\mathrm{Gene}\;\mathrm{Expression}=\mathrm{Mesor}+\mathrm{Amplitude}\;\times \;\cos(\frac{2\pi }{24h}\;\times \;(\mathrm{Time}\;-\;\mathrm{Acrophase}))$$

We used the “cosinor2” package in R to assess the amplitude and phase shifts (acrophase) of circadian expression patterns for each treatment condition, providing insight into the effects of tested conditions on rhythmicity characteristics. We used the rhythm detection test (also called the zero-amplitude test) as described before [35], with the *cosinor.detect* function, in order to validate the presence of significant rhythmicity in our model and therefore the applicability of the package in our dataset. This test is particularly valuable in circadian rhythm research, where it helps to ascertain whether observed fluctuations in variables like gene expression are indeed rhythmic or merely random variations [36]. We employed the *cosinor.poptests* to perform differential rhythmicity analysis, performing pairwise comparisons (control vs. UVB; control vs. D_3_), since the aforementioned function can only compare two populations at a time. For significant effects on the acrophase, the amplitude effect could not be interpreted from this analysis, so group comparisons in regard to the amplitudes were performed using unpaired *t*-tests, as similarly determined before [37].

#### 2.5.2. Linear Mixed-Effects Modeling with “nlme”

To complement the cosinor analysis and account for the hierarchical structure of our data, which include repeated measurements over time, we utilized a linear mixed-effects model (MELM). The model is defined as
Gene Expression ~ Condition × (cos(2π × Time/24) + sin(2π × Time/24)), 
with a random intercept for each repetition (random = ~1|Repetition). This approach allowed us to evaluate the fixed effects (treatment conditions) on the mean expression levels and the rhythmic components modeled by the cosine and sine terms.

#### 2.5.3. Two-Way Repeated Measures ANOVA

We further used mixed analysis of variance (ANOVA) to assess the statistical significance performed with SPSS v26 (IBM, Armonk, NY, USA) in regard to the overall gene expression. In regard to the first experiment, this was performed to assess the overall differences in gene expression without fitting them to a specific time-based model (like the sinusoidal pattern of a cosinor analysis). For different conditions of the same cell samples (the first experiment) we used repeated measures ANOVA with treatment conditions as the within-subjects factor and—where multiple time points were measured—time as the between-subjects factor. This distinction was made to account for the differences that occurred in HaCaT keratinocytes even after a short amount of time, owing to a short population doubling time of only around 28 h [26,38] and since their expression has been shown to differentiate based on growth and confluence [39]. In essence, while the UVB and D3 treatments were controlled and applied uniformly, the passage of time brought about uncontrolled biological variations that justified a more conservative statistical approach. Similarly, for different cells (the second experiment with NHEK, HaCaT, and SCL-1), the cell type was regarded as a between-subjects factor. We analyzed the interaction effects, meaning the dependency of the effect of one variable on the effect of another, with post hoc Tukey HSD. The mean differences were considered to be significant when *p* ≤ 0.05. For illustrations we used either −ΔCt values (as they were inversely correlated to gene expression) or 2^−ΔΔCt^ based on which provided a clearer understanding of the results. Nevertheless, in all statistical tests on gene expression we systematically used only ΔCt values.

In summary, the cosinor2-package approach was employed for its superiority in assessing the presence of rhythmicity and differential rhythmicity analyses under the tested conditions, while the MELM approach through the nlme package complemented this approach especially in regard to the effects on the overall gene expression due to its ability to account for both fixed and random effects. Additionally, with this model we could conduct analyses on effects involving datasets (Per2 gene; see below) that did not conform to the assumptions of the classical cosinor method (through the cosinor2 approach). Last, we chose to include our repeated measures ANOVA analysis to gain a simplified view of the treatment effects as this approach is commonly used in similar studies.

## 3. Results

### 3.1. Spontaneously Immortalized Human Epidermal Cells (HaCaT) Express Two Major CCGs (Bmal1 and Per2), with Bmal1 Showing a Robust Circadian Rhythm in Our In Vitro Model

Conducting the rhythm detection (zero-amplitude) test we found a significant effect for Bmal1 under the control (*p* = 0.002), UVB (*p* = 0.027), and D_3_ (*p* = 0.028) conditions, while for UVB+D_3_, no statistically significant rhythmicity was suggested. For Per2, circadian rhythmicity could not be found both in the control and under all tested treatment conditions (see Table 2). This also suggested that while our data of Bmal1 expression could be analyzed through the cosinor method, the interpretation of such an analysis with the Per2 data should be conducted with caution. The MELM analysis on Bmal1 gene expression revealed a significant effect of both the cosine term (*p* < 0.001) and the sine term (*p* = 0.037) circadian components, further validating our results of a robust circadian rhythm in Bmal1. In comparison, for Per2 expression a similar effect could not be evidenced (the cosine and sine terms were not statistically significant; *p* > 0.05 in both cases), which is in line with the Per2 pattern in our experimental model lacking significant oscillatory behavior.

### 3.2. Treatment of HaCaT Cells with UV-B And/Or 1,25(OH)_2_D_3_ Exerts Differential Effects on Expression of Bmal1 and Per2, with Both UVB and 1,25(OH)_2_D_3_ Significantly Modulating Bmal1 Rhythmicity Characteristics In Vitro

The cosinor analysis with the “cosinor2” package for Bmal1 showed a significant effect for 1,25(OH)_2_D_3_ (*p* = 0.001) indicating a suppressive effect over time. In regard to rhythmical characteristics, both UVB and 1,25(OH)_2_D_3_ indicated significant effects on the acrophase (*p* = 0.001 and *p* = 0.002, respectively), suggesting a phase shift of the peak, as also seen in Figure 1. Due to the significant effect on the acrophase, the amplitude was interpreted through unpaired *t*-tests (control vs. UVB and control vs. D_3_), which both failed to show a significant effect (*p* > 0.05). Per2 was excluded from this part of the analysis, as explained before. In regard to the MELM, a significant increasing effect of UVB (*p* = 0.021 with a coefficient of 0.73) and of the UVB + D_3_ combination (*p* = 0.02 with a coefficient of 0.74), but not of D_3_, could be found on the overall Bmal1 gene expression. On the other hand the model suggested a significant decreasing effect of D_3_ (*p* = 0.018 with a coefficient of −0.68) and a paradoxical increasing effect of UVB + D_3_ (*p* = 0.038 with a coefficient of 0.60), but not of UVB (*p* = 0.15), for the overall Per2 gene expression. In the ANOVA analyzing expression irrespective of time factor, we again saw a significant increasing effect of UVB on Bmal1 gene expression (*p* < 0.001) but also on Per2 (*p* < 0.001). A significant decreasing effect of D_3_ treatment on Per2 expression could also not be evidenced in this kind of analysis. In terms of the overall gene expression, we found an interaction effect of UVB*D_3_ (*p* = 0.014) for PER2. We analyzed this further by repeating testing of the effect of 1,25(OH)_2_D_3_ separately on the UVB-treated [UVB(+)] and UVB-non-treated [UVB(−)] samples. For UVB(+) we found *p* = 0.12, indicating no significance, but for UVB(−) we found *p* < 0.001 (Figure 2C,D).

In summary, the Cosinor analysis of Bmal1 expression indicated a significant phase shift after both UVB and D_3_ treatment, while the MELM model captured a significant effect on the overall gene expression of UVB on Bmal1 and of D_3_ on Per2. ANOVA validated the increasing effect of UVB, extending it to Per2 as well, while also suggesting a probable D_3_ effect there too.

### 3.3. Differential Expression and UV-B Effect on BMAL1 in NHEK, HaCaT, and SCL-1 Cells

The expression of BMAL1 was almost completely suppressed in SCL-1 and relatively reduced in HaCaT cells, as compared with NHEK. A strong statistical significance was namely found between NHEK/HaCaT and SCL-1 (*p* < 0.001 and *p* < 0.01, respectively) but not between NHEK and HaCaT (see Figure 3). We additionally found a significant UVB*cell type interaction effect (*p* < 0.001), which when further analyzed showed a relevant differential effect of UVB on SCL-1 cells, compared with NHEK and HaCaT (*p* < 0.001 in both cases). These effects were not seen for PER2, whose overall expression was only marginally reduced in HaCaT and SCL-1 as compared with NHEK.

### 3.4. UV-B Radiation-Induced Cellular Toxicity Is Only Marginally Altered by Co-Treatment with 1,25(OH)_2_D_3_, VDRi, and/or AhRi

The treatment of HaCaT cells with UV-B resulted in a significant increase (*p* = 0.024) in cellular toxicity. We found no significant effect on the toxicity of co-treatment with 1,25(OH)_2_D_3_, VDRi, and/or AhRi.

## 4. Discussion

### 4.1. Implication of UV-B-Regulated Expression of BMAL1 and PER2 in HaCaT Keratinocytes

As mentioned in the Introduction, evidence about the regulatory role of CCGs in the response mechanisms against UV-B radiation is plentiful. Knowledge regarding the direct effects of UVB radiation on CCG expression, on the other hand, is limited. In fact, since the study by Kawara et al. in 2002 [40], only one in vitro [41] and one in vivo [42] study have specifically investigated the effects of UVB on CCGs in keratinocytes and not vice versa. None of those studies in fact elaborated on the time-specific effects of UVB radiation. The study by Kawara et al. [40] found an initial suppression of all CCGs, which was restored in the first 24 h following treatment. Although no information about statistical analysis was offered, nor a control condition with which to compare the expression curves, the initial suppression of CCGs after treatment could be indicative of a relative decrease in overall gene expression, which could contradict our results. Furthermore, we are the first to our knowledge to observe a significant phase shift following UVB radiation in cutaneous tissues. Restricted feeding (RF) in a murine study also induced a phase shift in the cutaneous circadian clock, which directly influenced expression of the Xpa (key gene in DNA repair as discussed before), thereby altering sensitivity against UVB [43]. Cellular toxicity 24 h following UVB treatment in HaCaT keratinocytes was significantly increased in our model (*p* = 0.024), which is in accordance with previous research [44] and could possibly be linked to the observed CCG induction. At the same time, UVB has a long-known leading role in skin cancer, with the main known mechanism involving the accumulation of DNA damage photoproducts [16], which is heavily dependent for its restoration on a circadian clock-controlled repair system [13]. It can be speculated whether the effects of UV-B radiation on the expression of CCGs may be of relevance for skin photocarcinogenesis. This hypothesis is supported by the promotion of cancer cell invasion in Bmal1-depleted cells and its reduction when Bmal1 was overexpressed in a similar study [45].

Although no study has to our knowledge explicitly researched the circadian rhythms of cutaneous squamous cell carcinoma (cSCC), there have been investigations of other forms of cancer in which significant disruption of CCGs was also reported, with examples including head and neck SCC (downregulation) [46], gastric cancer (upregulation) [47], breast cancer (variable depending on type of cancer) [48], etc. The exact etiology of why several cancers indicated altered circadian rhythms is not yet fully understood. While our current study did not directly link the increased expression of CCGs to specific carcinogenic pathways (like, e.g., p53), the observed cellular toxicity and the findings from our previous study on VDR and AHR pathway modulation suggest potential pathways through which UVB-induced CCG alterations could contribute to carcinogenesis. Our previous research indicated differential regulation of VDR and AHR target genes by UVB and 1,25(OH)_2_D_3_ in HaCaT and SCL-1 cells [49]. This suggested a complex interplay between UVB-induced pathways and the regulatory network involving VDR and AHR, which are known to influence carcinogenic processes and especially for AHR are directly linked to the circadian clock circuitry [50]. Future studies are planned to explicitly investigate these connections, particularly focusing on the mechanistic links between CCG dysregulation and key cancer pathways such as p53. UVB has been shown to directly influence CCG expression in epidermal skin and subcutaneous adipose tissue in vivo [42], although the exact rhythmicity effects over time and the consequences of those effects need further research.

Last, due to CCGs’ direct link to the cell cycle and cell senescence [51] it is possible that these effects observed here on CCG expression reflect the cellular coping mechanisms in signaling damage and ridding the organism of damaged—with dysregulated circadian rhythms—cells. In other words, the overexpression of CCGs could be a response to UVB and not its direct effect and therefore a cancer-protecting characteristic, rather than a sign of increased carcinogenicity.

### 4.2. Treatment of HaCaT Cells with UV-B and/or 1,25(OH)_2_D_3_ Exerts Differential Effects on Expression of Bmal1 and Per2 In Vitro Indicating Alternative Mechanisms of Action

We investigated the hypothesis, whether the circadian clock-regulating effects of UV-B may be mediated via vitamin D-dependent signaling. UV-B is essential for the cutaneous synthesis of vitamin D_3_, a molecule with multiple important biologic functions. The pathway starting in keratinocytes and leading up to the formation of its active form 1,25(OH)_2_D_3_ (calcitriol), which exerts its endocrine effects, has been described before [52]. This hormonally active form of D_3_ mediates its numerous functions mainly by binding to the vitamin D receptors (VDR) in VDR-positive target tissues [52]. Interestingly, keratinocytes, unlike some other cell types (e.g., fibroblasts), are capable of fully synthesizing 1,25(OH)_2_D_3_ locally from 7-DHC for autocrine/paracrine effects. An interplay between VDR and p53 has recently come to light [52,53], while at the same time, multiple lines of evidence suggest a bilateral regulation of p53 and the CCGs [21,54,55,56,57,58].

Gutierrez-Morreal et al. [59] demonstrated that culturing adipose-derived stem cells (ADSCs) with 1,25(OH)_2_D_3_ (both continuous and spiked treatment modalities) resulted in a significant synchronization of the expression of the CCGs Bmal1 and Per2, meaning an increase in their amplitude, but no statistics regarding the effects on other characteristics including overall gene expression were provided. Moreover, in patients receiving dental implants, Mengatto et al. [60] found differential expression of several CCGs, comparing “normal” controls with vitamin D_3_-deficient patients. Additionally, evidence suggests a regulatory role of the intestinal circadian system in skeletal bone homeostasis. The circadian CLOCK protein was reported to be interacting with VDR: this interaction was BMAL1-dependent, thus enhancing its transcriptional activity in a rhythmic way [61].

To best of our knowledge, we are the first to study the effect of 1,25(OH)_2_D_3_ (and its interaction with UV-B) on CCGs expression in skin cells. We here report differential, in part opposite, effects of UV-B radiation and 1,25(OH)_2_D_3_ on the expression of Bmal1 and Per2. While after treatment with 1,25(OH)_2_D_3_, expression of Per2 was significantly decreased, UV-B-treatment enhanced its expression, which was only of statistical significance when the time factor was not taken into account. At the same time, D_3_ also induced a phase shift of Bmal1, in a similar way but to a lesser extent than UVB (see Figure 1). Especially the effect of D_3_ on Per2 would be interesting to further investigate in the context of the studied p53/VDR interplay [62] as p53 has been found to directly bind on Per2 and regulate its expression [54]. The p53 gene is regarded as “guardian of the genome”, is critically involved in processes that govern DNA damage, and acts as a major tumor suppressor gene in many cancers, including non-melanoma skin cancer. When DNA damage is recognized, p53—dependent on whether repair is possible—can either arrest the cell cycle to encourage repair or alternatively induce apoptosis, thereby obstructing the accumulation of damage in cells and their consecutive mutagenesis [63]. Whether the lack of modulation of the UV-B-induced cellular toxicity by vitamin D derivatives in HaCaT that we here report is linked to the altered p53 status (p53mut) in these cells also deserves further analysis.

These observations showing unparallel and partly opposite effects of UVB and D_3_ on CCG expression do not support the concept that the circadian clock-regulating effects of UV-B in skin may be mediated via vitamin D. It should be noted that D_3_ has been many times investigated for its photoprotective effects. Despite not representing a classical antioxidant, it has been shown to antagonize UV-induced oxidative stress. In skin cells, this effect is associated with a reduction in UV-induced DNA damage, an increase in p53 levels (that have been linked with DNA repair and reduction in reactive oxygen species (ROS)), inhibition of stress-activated kinases, and the induction of metallothioneins [64]. Moreover, 1,25(OH)_2_D_3_ decreases UV-induced NO-mediated nitrosylation of DNA repair enzymes. Other functions of vitamin D_3_ in favor of NER and the repair of DNA damage overall are nevertheless also suspected [65]. Furthermore, 1,25(OH)_2_D_3_ has exhibited anti- proliferative and pro-differentiation effects on BCC and SCC lines in vitro. Several lines of evidence indicate that the VDR acts as a tumor suppressor, interacting with the p53 family of proteins (p53/p63/p73 proteins). Some of the effects exerted by the VDR/p53 interaction include a reduction in CPDs and NO-products and the regulation of the murine double minute (MDM2) gene—a gene that negatively affects p53 expression. Interestingly, the combination of UVB + D_3_ in both Bmal1 and Per2 showed effects parallel to those of UVB. Whether this was part of a more complex interplay between the two factors or an effect dependent on the balance of doses of each condition needs further investigation. More importantly, whether the observed effects of lone D_3_ constitute a novel photoprotective property remains to be further explored.

### 4.3. Differential Expression of BMAL1 in NHEK, HaCaT and SCL-1 Cells Indicates Disruption of Circadian Rhythm during Skin Photocarcinogenesis

To shed further light in skin photocarcinogenesis, we used the types of skin cells that differ in their p53 status: NHEK(p53wt), HaCaT(p53mut), and SCL-1 (no p53 protein expressed) cells as an in vitro model for normal skin, actinic keratoses, and cutaneous squamous cell carcinomas respectively. Expression of Bmal1 was almost completely suppressed in SCL-1 and marginally reduced in HaCaT cells, as compared with NHEK. It should be noted that drawing general conclusions without introducing a time factor should be conducted with caution. Nevertheless, these findings provide first evidence that supports a probable disruption of circadian clocks during skin photocarcinogenesis but also showcase how disrupted rhythms could be translated into different responses to stimuli, like in this case, UVB. Exploiting such differences could be of high clinical relevance in creating novel therapeutic applications targeting the genes of the CC, and therefore, further research is necessary.

### 4.4. Limitations

Lack of Cell Synchronization: Not synchronizing the cells prior to treatment may have resulted in varied circadian phases among the cell population, potentially affecting the interpretation of gene expression changes. This could also have influenced the lack of a robust circadian rhythm in Per2 expression, for which reasons the corresponding effects found upon it should be interpreted with caution.Intercell differences: The experiment investigating intercell differences was conducted only at one point, which made generalized conclusions difficult. Further studies should explore such differences evidenced here introducing the time factor through multiple measurements.Mechanistic Insights: The study primarily focused on expression levels without delving deeply into the underlying molecular mechanisms or the functional consequences of these changes.

## 5. Conclusions

In conclusion, we demonstrated a pattern of circadian rhythm in the expression of two major CCGs (Bmal1 and Per2) in human skin cells (HaCaT) in vitro. This pattern was differentially modulated both by vitamin D derivatives and by UV-B, and it varied greatly when we compared cells representing different stages of skin photocarcinogenesis. While these findings did not support the hypothesis that the circadian clock-regulating effects of UV-B may be mediated via vitamin D, they did support the concept that disturbance of the CC may be of relevance for skin photocarcinogenesis, while potentially introducing a novel pathway in which vitamin D exerts its photoprotective properties.

## Figures and Tables

**Figure 1 nutrients-16-00254-f001:**
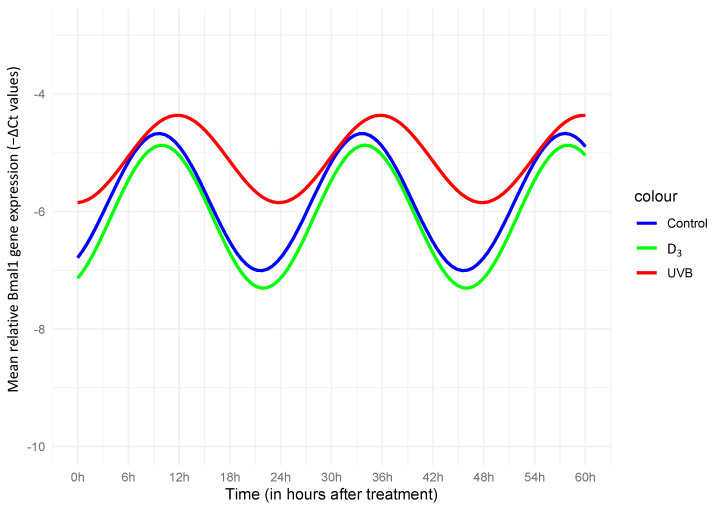
Cosinor plot with the fitting curve, for −ΔCt values of Bmal1, normalized to GADPH. A significant effect of both UVB (*p* = 0.001) and D_3_ (*p* = 0.001) on the acrophase indicated a phase shift, meaning the timing of the peak high of the oscillation curve happened significantly later in comparison with that of the control curve. A significant effect in the amplitude was not suggested under either condition. As shown in the illustration, both a relevant phase shift and an amplitude decrease under UVB could be assessed, while for D_3_, the effects seemed rather insignificant, closely following the estimated rhythmic pattern of the control.

**Figure 2 nutrients-16-00254-f002:**
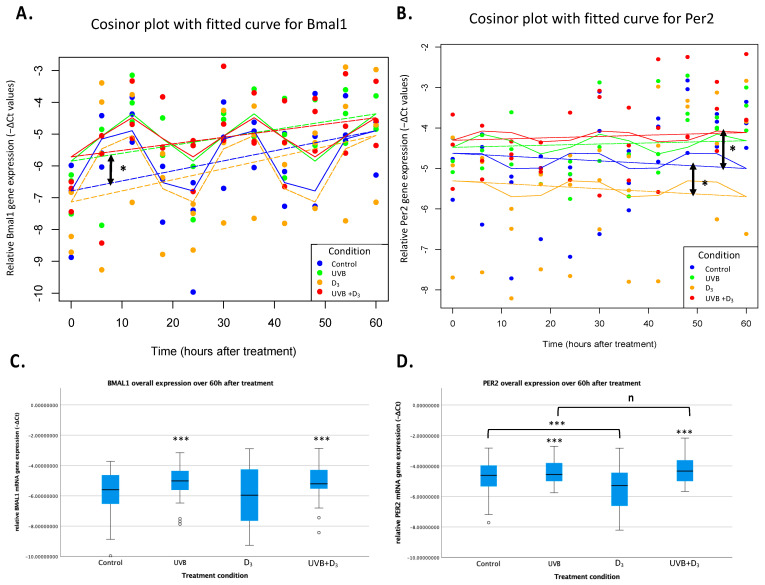
Circadian rhythm of relative mRNA gene expression of Bmal1 (**A**,**C**) and Per2 (**B**,**D**) in HaCaT keratinocytes (−ΔCt values normalized to the GADPH house-keeping gene). Mixed-effects cosinor plot for Bmal1 (**A**) and Per2 (**B**). The curve represents the best-fit sinusoidal curve that models the circadian rhythmicity of gene expression for Bmal1 and Per2. It was based on the combined cos(2π × Time/24) and sin(2π × Time/24) terms from the mixed-effects model, reflecting the predicted rhythmic pattern over time. The horizontal dashed line indicates the mesor, which is the midline estimating statistic of rhythm. The mesor represents the average level of the rhythmic function over a complete cycle, serving as a baseline around which the oscillations occur. There was a significant increase in Bmal1 expression (*p* = 0.021) after UVB treatment, as well as a decrease in Per2 (*p* < 0.018) after D_3_ supplementation, as indicated by the asterisks and double arrows. While UVB did not significantly increase Per2 expression, its combination with UVB mediated a significantly (*p* = 0.038) opposite (increasing) effect to that of D_3_ alone. This could reflect a probable stronger increasing effect of UVB, which could not be found in this specific model, or a more complex interplay between UVB and D_3_. In (**C**,**D**) the illustration of analysis of overall gene expression irrespective of the time factor (two-way repeated measures ANOVA). UV-B treatment increased the overall expression of both Bmal1 and Per2 (*p* < 0.001 (**C**,**D**)). Treatment with 1,25(OH)_2_D_3_, on the other hand, showed a significant effect on non-UVB-treated samples but not in UVB-treated ones. The decreasing effect of lone D_3_ treatment found in our MELM analysis was here absent. Different effects of UVB and 1,25(OH)_2_D_3_ in both cases speak nevertheless against the 1,25(OH)_2_D_3_ being a mediator of UVB-induced effects (n: *p* > 0.05, *: *p* < 0.05, ***: *p* < 0.001).

**Figure 3 nutrients-16-00254-f003:**
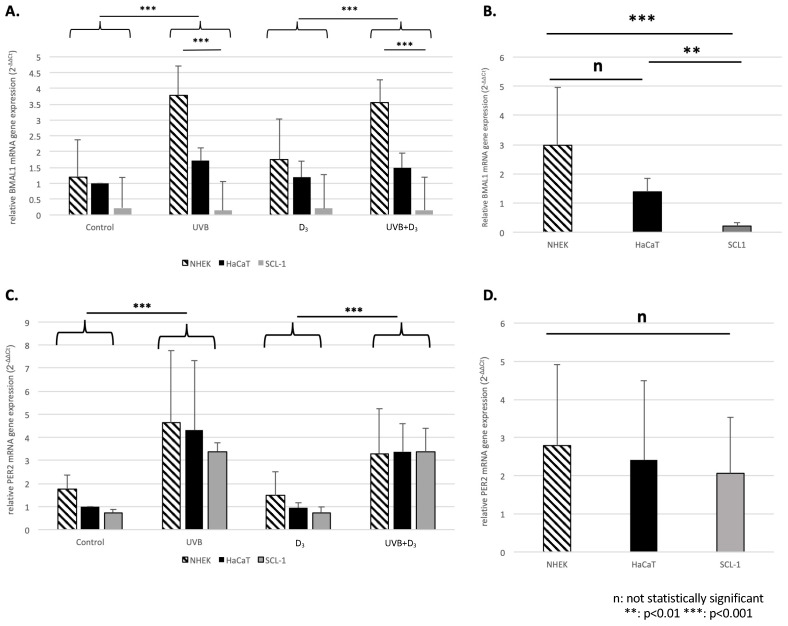
Relative mRNA gene expression of BMAL1 (**A**,**B**) and Per2 (**C**,**D**) 24 h following treatment in NHEK, HaCaT, and SCL-1 cells normalized to GADPH. The mean expression ± SD (in this instance, fold ratio 2^−ΔΔCt^ relative to the control condition of HaCaT for a better representation of the results) is illustrated for the different treatment conditions (**A**,**C**). For both BMAL1 (**A**) and Per2 (**C**) we saw a significant effect (*p* < 0.001) of UVB. For BMAL1 we also saw a differential effect of UVB between HaCaT/NHEK (*p* < 0.001; upregulation) and SCL-1 (downregulation), which was not evident for Per2. Additionally, an illustration of the overall expression of HaCaT, NHEK, and SCL-1 (B: BMAL1, D:Per2) indicated a significant differential expression of BMAL1 between HaCaT/SCL-1 (*p* = 0.003) and NHEK/SCL-1 (*p* < 0.001) but no difference between HaCaT/NHEK, as well no differential expression regarding Per2.

**Table 1 nutrients-16-00254-t001:** Combinations of treatments used in HaCaT keratinocytes for LDH toxicity level assessment. The sampling of the medium happened 24 h following the respective treatment. A total of 16 different combinations of the four illustrated treatment options: UVB, 1,25(OH)_2_D_3_, AhR-inhibitor (CH223191), and VDR-inhibitor [25(OH)_2_D_3_]. Levels of LDH toxicity were assessed relative to the LDH measured in a 1% Triton X-100 treated sample, which lysed cells and represented the maximum amount of LDH enzyme activity.

	Sample 1	Sample 2	Sample 3	Sample 4	Sample 5	Sample 6	Sample 7	Sample 8	Sample 9	Sample 10	Sample 11	Sample 12	Sample 13	Sample 14	Sample 15	Sample 16
UVB	-	+	+	+	+	-	-	-	-	-	+	-	-	+	+	+
D_3_	-	-	+	+	+	+	-	-	-	+	-	+	+	-	-	+
AhR-i	-	-	-	+	+	+	+	-	+	+	-	-	-	+	+	-
VDR-i	-	-	-	-	+	+	+	+	-	-	+	+	-	-	+	+

**Table 2 nutrients-16-00254-t002:** Rhythm detection (zero-amplitude) test for Bmal1 and Per2 ΔCt values, GADPH normalized, under each condition, to assess the presence of rhythmicity. For Bmal1 a robust circadian rhythm was found and remained consistent in the control, UVB, and D_3_ conditions (*p* < 0.05), while its significance was only marginal for the combination of UVB + D_3_. For Per2, no circadian rhythmicity could be proven both in the control and under all tested combination conditions. (*df*1: degrees of freedom in numerator, *df*2: degrees of freedom in denominator). Rhythmicity characteristics (MESOR, amplitude, and acrophase) for Bmal1 extracted by the cosinor model to describe fitted curves as illustrated in Figure 1. For its interpretation, it should be noted that ΔCt values were inversely related to gene expression.

Gene	Condition	F-Statistic	*df*1	*df*2	*p*-Value	Rhythmicity Characteristics		
Bmal1	Control	7.217496	2	28	0.002965341	MESOR	Amplitude	Acrophase
						5.20925	1.707255	−1.601187
Bmal1	UVB	4.114259	2	28	0.02713342	MESOR	Amplitude	Acrophase
						5.457692	1.078301	−4.863345
Bmal1	D_3_	4.054122	2	28	0.0284265	MESOR	Amplitude	Acrophase
						6.722672	2.43367	−4.159739
Bmal1	UVB + D_3_	2.248605	2	28	0.1242703	omitted due to non-conformity with the cosinor method		
Per2	Control	2.541108	2	28	0.09680284	omitted due to non-conformity with the cosinor method		
Per2	UVB	2.152137	2	28	0.1350743	omitted due to non-conformity with the cosinor method		
Per2	D_3_	1.630983	2	28	0.2137886	omitted due to non-conformity with the cosinor method		
Per2	UVB + D_3_	1.167377	2	28	0.3258711	omitted due to non-conformity with the cosinor method		

## Data Availability

Data is contained within the article and Appendix A.

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
