# Peer review of "Differential Regulation of Circadian Clock Genes by UV-B Radiation and 1,25-Dihydroxyvitamin D: A Pilot Study during Different Stages of Skin Photocarcinogenesis"

_nutrients, 2024, doi:10.3390/nu16020254_

Round 1
Reviewer 1 Report
Comments and Suggestions for Authors
The authors present the differential expression of circadin clock genes in different conditions.
Even though the authors apply basic statistical tests, such as ANOVA and Tukey HSD test, I suggest using specific methods devoted to the analysis of circadian data to analyse the circadian profiles of observed genes as well as differences among these profiles. These methods include Cosinor (10.1186/1742-4682-11-16), JTK_Cycle (10.1177/0748730410379711) and RAIN (10.1177/0748730414553029). Software packages with the implementations of circadian data analysis methods that can be used straightforwardly and only require basic programming skills are widely available and relatively easy to use, e.g., see CosinorPy - Python (10.1186/s12859-020-03830-w, https://github.com/mmoskon/CosinorPy) or Cosinor2 - R (https://cran.r-project.org/web/packages/cosinor2/index.html).
Is the rhythmicity after specific treatment condition disrupted? How?
The authors only observe/compare the overall expression of BMAL1 and PER2 and/or expression in a single time point, which can lead to misleading conclusions. I suggest that the authors investigate the time-dependant circadian profiles of the available data (e.g., rhythm amplitudes, and acropahse). These can also be use to investigate differential rhythmicity among different treatments and significance of changes among them.
Overall, I strongly encourage the authors to perform the analysis of the data as timeseries data.
Figure 1 (A, B, C, D) should also include error bars.
Author Response
First of all we would like to thank you for your constructive feedback and the idea of analysing our data as time series. We performed the cosinor analysis, with the use of R programming language as suggested, employing the "nlme" package, which provided us with another dimension, as many effects that were not found in our first analysis were now evidenced. Because of this I removed parts of the descriptive analysis on how the expression curves look and focused more on the numbers, as this seems more systematic. We chose to still include the part of the analysis referring to overall gene expression with the use of ANOVA, as it could still be relevant especially in conjuction with the effects on elements of rhythmicity, as especially some conditions have an effect on overall expression when time-factor is taken into account and some others the other way round. We also added the code and output used for both testing overall rhythmicity and of the effect of conditions on it as supplemental material.
Reviewer 2 Report
Comments and Suggestions for Authors
In this manuscript, the authors tested a hypothesis on the contribution of UV-B and vitamin D3 on the circadian clock. This work extended the finding that the circadian clock plays a crucial role in regulating cellular processes beyond the sleep-wake cycle in peripheral tissues, specifically, the circadian clock’s potential involvement in UV-B induced skin cancer. Using RT-qPCR, the study examined the expression of two core clock genes (CCGs; Bmal1 and Per2) in HaCaT cells across different time intervals (0-60 hours). They investigated the expression of these genes in keratinocytes at various stages of skin photocarcinogenesis: normal (NHEK; p53 wild type), precancerous (HaCaT keratinocytes; mutated p53 status), and malignant (SCL-1; p53 null status) cells. This analysis was conducted 12 hours after treatment with or without 1,25-dihydroxyvitamin D (D3) and/or UV-B. The result demonstrated that there was a circadian rhythm pattern in the expression of Bmal1 and Per2 in HaCaT cells, with an overall increase in gene expression following UV-B exposure. Varied effects on Bmal1 and Per2 expression were observed depending on the treatment approach (UV-B and/or D3) or cell types (NHEK, HaCaT, and SCL-1 cells). From these results, the authors claimed that (1) their results support the independently functioning timekeeping system in the human skin, (2) this system can be regulated by UV-B and disrupted during skin photocarcinogenesis, and (3) the circadian rhythm pattern was differentially affected by UV-B treatment compared to D3 treatment, contrary to previous literature.
General comments
The manuscript is clear and well-structured. However, it suffers from many flaws in experimental design. The experimental results are too simplified for the conclusion the authors were trying to draw. Specifically, in the methods, the properties of the intrinsic circadian rhythms (phase and period) of the cell samples are not specified prior to and by the time of UV-B treatment, which could invalidate the comparison of the effect of UV-B light between different cell types (as the cells could be exposed to UV light at different phases). Even within the HaCaT cell line, the overall increase of Bmal1 and Per2 gene expression after UV-B treatment does not necessarily mean that these alterations could lead to the disruption of the circadian clock system and photocarcinogenesis, as the phase did not change (only the overall gene expression increased). Moreover, it was also mentioned by the authors that this effect is only temporary and the CCG expression recovered to a baseline 12-20h after UV-B treatment (Figure 1C). Therefore, the evidence provided by the manuscript is not sufficient for their conclusion. There was missing information on the consequences of the UV-B-induced increased expression of CCGs to pathways related to carcinogenesis (e.g., p53).
Specific comments
In the Introduction, even though the authors provide background information about circadian rhythms, and their connection with skin cells, there is not enough information regarding how it can interact with photocardinogenesis pathways.
In the Methods, the authors should include references to justify the dosages of UV-B and 1,25(OH2)D3 treatment required for the experiment.
In the results, D3 seems to have an opposing effect compared to UV-B, which has not been mentioned or discussed by the authors.
Some specific comments below:
Line 24 onwards: For transcripts, the convention is to write as Bmal1, not BMAL1, which is reserved for proteins.
Lines 40-45: Although this is a general introduction, it bears too much similarity to Lubov JE et al (2021) Int J Mol Sci; PMID: 34204077.
Line 50 & 68: The acronym CCGs has been doubly defined. CCGs most often stands for clock controlled genes, or may be core clock genes.
Line 180: Similar to the treatment condition, measured time should also be considered within-subject factor as the same groups of cells are measured across time.
Line 183 & 185: It is not clear whether Tukey’s HSD or Bonferroni was used to correct for multiple comparisons.
Line 187 & 188: It is not justifiable if -ΔCt or 2-ΔΔCt could lead to differences in statistical significance. If not, the results should be illustrated consistently.
Line 222: typo: p’<0,001
There were no ethics statements or data availability statements for this manuscript.
Author Response
Please see the attachment.
We want to thank you for your constructive feedback. Please see the attachment for a point-to-point answer to your comments. Answers have been concluded as part of the revisions, where applicable. In the revised manuscript, you will find a revision of the Introduction (as suggested by you), an addition in Methodology due to a new analysis of our data and subsequent changes in the Results, Discussion and Abstract.

Round 2
Reviewer 1 Report
Comments and Suggestions for Authors
Even though the authors now perform the timeseries analysis based on the cosinor model, they should perform the zero amplitude test using this model (see Cosinor2 or CosinorPy implementations). The visualisations of the models could also be improved. Moreover, more relevant references as suggested in the first revision should be cited.
I also encouarge the authors to uses the cosinor model to perform the differential rhythmicty analysis (as also supported by Cosinor2 or CosinorPy).
Author Response
A cosinor analysis war also performed using the cosinor2 package, with which a zero-amplitude test was also conducted as suggested. We combined results found from our previous analysis into a three-parts comprehensive analysis involving all three approaches, trying to showcase each method's advantages. The effects seen for Bmal1 are understandably more sound and are therefore highlighted as such. We still chose to report all findings (even the ones from ANOVA), as our goal with this pilot study was to gain some first insights into underexplored areas of dermatological research and pave the way for further, more comprehensive studies on the matter. Notably, in the literature no other study has studied effects of any condition on HaCaT keratinocytes in such a systematic way, even if especially the connection between CCG and UVB has been extensively investigated.
We also added further visualisation, as suggested, especially owing to the now more extensive analysis. We have cited more citations relevant to the statistics, to parts of the discussion, while in our previews revision we also provided further background, and therefore resources, into our introduction.
Reviewer 2 Report
Comments and Suggestions for Authors
Many issues are addressed in this version and concerns have been cleared. In the Supplementary File, the names of the genes are randomly in uppercase or lowercase. Please correct them.
Author Response
Thanks for your feedback. The data in the supplementary file have been corrected.
As part of the revisions requested from the other reviewer, we deepened our statistical analysis, finding some further effects on rhythmicity. For this reason we added/modified some parts of the discussion.